# Can Artificial Intelligence Aid Diagnosis by Teleguided Point-of-Care Ultrasound? A Pilot Study for Evaluating a Novel Computer Algorithm for COVID-19 Diagnosis Using Lung Ultrasound

Laith R. Sultan [1,*], Allison Haertter [2], Maryam Al-Hasani [3], George Demiris [4], Theodore W. Cary [3], Yale Tung-Chen [5] and Chandra M. Sehgal [3]

1 Department of Radiology, Children's Hospital of Philadelphia, Philadelphia, PA 19104, USA
2 Radiation Oncology Department, University of Pennsylvania, Philadelphia, PA 19104, USA; ahaertter@nyproton.com
3 Ultrasound Research Lab, Department of Radiology, University of Pennsylvania, Philadelphia, PA 19103, USA; maryam.al-hasani@pennmedicine.upenn.edu (M.A.-H.); tedcary@gmail.com (T.W.C.); chandra.sehgal@pennmedicine.upenn.edu (C.M.S.)
4 Informatics Division of the Department of Biostatistics, Epidemiology and Informatics, University of Pennsylvania, Philadelphia, PA 19104, USA; gdemiris@nursing.upenn.edu
5 Emergency Medicine Department, La Madrida Hospital, 28006 Madrid, Spain; yale.tung.chen@gmail.com
* Correspondence: sultanl@chop.edu

**Abstract:** With the 2019 coronavirus disease (COVID-19) pandemic, there is an increasing demand for remote monitoring technologies to reduce patient and provider exposure. One field that has an increasing potential is teleguided ultrasound, where telemedicine and point-of-care ultrasound (POCUS) merge to create this new scope. Teleguided POCUS can minimize staff exposure while preserving patient safety and oversight during bedside procedures. In this paper, we propose the use of teleguided POCUS supported by AI technologies for the remote monitoring of COVID-19 patients by non-experienced personnel including self-monitoring by the patients themselves. Our hypothesis is that AI technologies can facilitate the remote monitoring of COVID-19 patients through the utilization of POCUS devices, even when operated by individuals without formal medical training. In pursuit of this goal, we performed a pilot analysis to evaluate the performance of users with different clinical backgrounds using a computer-based system for COVID-19 detection using lung ultrasound. The purpose of the analysis was to emphasize the potential of the proposed AI technology for improving diagnostic performance, especially for users with less experience.

**Keywords:** artificial intelligence; point-of-care ultrasound; COVID-19; automated image analysis; telemedicine; augmented and virtual reality

## 1. Introduction

The coronavirus disease (COVID-19) pandemic has accelerated the adoption of technologies that facilitate patient care while reducing viral spread. Telemedicine is one of the technologies that expanded rapidly during the pandemic across a broad range of clinical practices [1–3]. It became the safest interactive system to maintain social distancing between patients and clinicians. One area of telemedicine that has an increasing potential is teleguided ultrasound, where telemedicine and point-of-care ultrasound (POCUS) merge to create this new scope [4,5]. Teleguided POCUS can minimize staff exposure while preserving patient safety and oversight during bedside procedures. As COVID-19 placed unprecedented pressure on healthcare services, care workers responsible for critically ill patients faced significant challenges. To meet such unprecedented demand for clinical services, additional workforce support will likely prove vital. This may involve utilizing other

potential healthcare workers alongside the existing staff. These potential new operators include those with alternate ultrasound (US) imaging experience [6,7] and those with no previous US imaging experience [8–11]. In this paper, we propose a pragmatic consideration of the remote monitoring of COVID-19 patients using teleguided POCUS, supported by AI technologies, by non-experienced personnel including the patients themselves. In pursuit of this goal, we performed a preliminary analysis to evaluate a novel computer algorithm for characterizing COVID-19 using POCUS. The performances of the algorithm users with different experiences ranging from non-expert to experienced user were compared. The primary aim of the tool we investigate in this study, is to demonstrate that similar AI-driven tools have the potential to significantly enhance the automated detection of both normal and abnormal findings, even when utilized by individuals with limited medical experience. Should these AI tools undergo validation in larger-scale studies, they could subsequently be integrated into POCUS devices. This integration would empower less experienced users to detect normal anatomical structures and abnormalities with a greater degree of automation and accuracy.

In essence, our research sets the stage for a promising future in which AI technologies play a pivotal role in enhancing the capabilities of healthcare professionals and even patients themselves in effectively monitoring and managing COVID-19 cases using remote teleguided POCUS.

## 2. Telemedicine in the Time of the Pandemic

The use of telemedicine during the pandemic for patients' management provided several additional advantages [1–3], such as reduced patient and employee exposure, increased safety and monitoring for patients with pre-existing medical conditions, and increased hospital personnel availability. Remote monitoring also decreased the total cost of care, with upwards of a 32% reduction in the cost of care, due to the shift to a value-based care model [4,5]. Many hospitals began such programs to support patients with COVID-19, in addition to many other conditions such as congestive heart failure, cancer, deep vein thromboses, and infections such as cellulitis and sepsis [12]. The U.S. military's health system actively used remote monitoring. Brooke Army Medical Center in San Antonio, for example, used a COVID-19 remote monitoring program in which patients are enrolled directly from hospital inpatient units or the emergency department prior to discharge. Kaiser Permanente, the Mayo Clinic, and Intermountain Healthcare in Utah scaled up and expanded their remote monitoring programs, where their virtual hospital is staffed 24/7 by remote monitoring technicians and has tele-nurses, tele-hospitalists, and other tele-advanced practice providers available depending on the patient's condition. Studies on the remote patient monitoring (RPM) of COVID-19-positive patients reported no fatalities occurring amongst the 244 patients involved [13,14]. These studies showed that RPM programs were able to reduce anxiety due to the uncertainties in, and constantly evolving nature of, guidelines amongst patients who have tested positive or are suspected of having COVID-19 [3,14] by providing patients with guidance on when symptoms are substantial enough to warrant hospital visits [3,13].

## 3. The Role of POCUS in the Management of COVID-19 Patients

Since its inception for military use in 1998, portable ultrasound or point-of-care ultrasound (POCUS) has grown in accessibility, useability, and technological sophistication [15], revolutionizing the accessibility of diagnostic imaging to resource-limited areas and emergency situations [15–17]. POCUS use has increased significantly during the pandemic for monitoring the inflammatory lung changes seen in COVID-19 patients [18–24]. This is facilitated in part by the typical sonographic characteristics of COVID-19-associated lung injury during disease progression and recovery [25–37]. Usually, POCUS is operated by experienced clinicians. The primary POCUS imaging technique is based on brightness mode (B-mode) imaging of the pleural lining and lung tissue [38–43]. Under normal circumstances, the tissue at the boundary between air and soft tissue or fluids is very

reflective, and tissues within the images are indistinguishable [42,43]; hence, with a highly aerated lung surface, ultrasound penetration beyond the pleural line is hindered. When the alveoli fill with fluid, which frequently occurs during infections, the interface becomes penetrable by ultrasound waves generating bright vertical artifacts called B-lines. Despite being well-described and utilized in clinical practice, the exact underlying cause of these vertical artifacts in the lungs remains unknown [42]. The cause of this phenomenon was initially attributed to reverberation events that occur when the ultrasound beam interacts with thickened interlobular septa; this is known as the "septal hypothesis". However, subsequent evidence has shown that this hypothesis is at least incomplete, as these artifacts have been observed in naturally collapsed healthy lungs and in synthetic phantoms. Yet, they do provide qualitative "visual" information about a range of pathologies from a clinical perspective [39], and the presence of dense multiple B-lines consistently corresponds to ground glass opacities (GGOs) seen on computed tomography (CT) scans, where GGOs represent a tomographic variation in the air/tissue ratio.

While POCUS technology has seen great advancements in accessibility, the issue that persists in both developing and developed areas of the world is a shortage of radiologists and sonographers who are able to perform the ultrasound imaging. It is this shortage of trained professionals that is driving the push for teleguided ultrasound useability.

## 4. Teleguided POCUS for Remote Monitoring of COVID-19 Patients

The feasibility of teleguided POCUS has been shown in several studies, the scope of which includes obstetrics [4,5,44,45] and abdominal, thoracic and lung, and trauma situations. Different techniques of teleguided US have been studied, including telerobotic and remote monitoring, with both types being deemed successful implementations regarding adequate image acquisition for diagnostic requirements, user comfort, patient acceptance, and increased accessibility to patients and medical systems.

Earlier works have already discussed the advantages, feasibility, and success of remote COVID-19 patient care, as well as the ability to monitor COVID-19 patients with POCUS. Studies have put forth specific guidelines to facilitate the potential implementation of lung ultrasound (LUS) in telemedicine applications. These guidelines concern the medical personnel who might require formal training and qualification, as well as the adoption of specific protocols for focused assessment [46]. Pivetta et al. (2020) [44] have already shown that an ultrasound-experienced individual can perform remote self-administration of POCUS with the aid of teleguidance. While self-performed ultrasound may sound improbable, it has already been performed, supporting the feasibility of the proposed model. There exists a report about a nurse who was exposed to COVID-19 and completed 16 days of self-performed POCUS to monitor their health progression with widely adapted protocols [46]. In general, the image quality achieved during self-performance was greatly enhanced once teleguidance was added to provide feedback on the imaging protocols used [47]. The study showed that self-performed POCUS is not only possible but can be successful in the detection of changes related to COVID-19 with ultrasound technology and teleguidance support. While this study was carried out by a nurse who had ultrasound imaging experience, there is evidence that novice users can produce diagnosable quality images with only a small amount of training [48–50]. The remote-monitored patients can be given a remotely administered didactic and practice session (or sessions) to provide instructions on how to correctly hold and use the ultrasound transducer and the locations in which they are to be targeting for imaging. POCUS scoring is in accordance with the protocol laid out by Soldati et al. (2020) [51].

Our proposed model is centered around teleguided POCUS for the remote monitoring of lung features specific to COVID-19. The aim is to enable untrained staff or even the patients themselves to perform the ultrasound scans under remote guidance. The largest limitation that can be foreseen is patient compliance, specifically amongst patients in the remotely monitored subgroup, who may be unwilling to self-perform daily POCUS imaging due to high discomfort levels, a fear of inadequate care, or the misunderstanding

of study protocols. Another potential concern is related to data protection in respect of ultrasound imaging and other acquired health records. This issue has been raised by other authors [5,50], and patients self-performing imaging could be subject to even higher risks.

The primary objective of this paper is to advance the proposed model by evaluating the practicality of novice users independently conducting daily POCUS scans, augmented by a computer-based analysis tool. This innovative approach seeks to address the inherent challenges associated with operator dependency and inconsistencies in imaging settings. By offering standardization in image acquisition, the computer-based tool aims to enhance the dependability and uniformity of POCUS outcomes. Ultimately, the implementation of such a system has the potential to refine the accuracy of lung-feature monitoring for individuals affected by COVID-19. This advancement would not only increase the accessibility of POCUS but also reduce the need for highly trained personnel to be present during the scanning process. By empowering untrained staff or even patients themselves to perform POCUS scans under remote supervision, this model strives to improve the efficiency and effectiveness of remote monitoring, contributing to better healthcare outcomes for COVID-19 patients.

## 5. AI Can Improve Teleguided POCUS Monitoring of COVID-19 Patients

For the thousands of patients fighting for their lives against this deadly disease and the health care providers who incur a constant risk of infection, the implementation of AI in healthcare can be a game changer. In particular, the use of AI applications can facilitate the teleguided POCUS monitoring of COVID-19 patients. The AI industry, as well as academia, has been developing algorithms for the detection of pathological changes related to COVID-19 [52–68]. Several image-based AI diagnostics have been built to detect COVID-19 changes by using different modes such as CT scans, X-rays, cough sounds, MRIs, and ultrasound, as well as clinical markers.

Implementing such tools in teleguided ultrasound would have the potential to facilitate remote testing and expand access. There are two ways in which AI can have an impact. The first is automated image analysis, particularly for people with less familiarity with imaging, to facilitate the effective use of the scanner. Users who are not experienced in lung imaging could progress to screening patients with high accuracy. Second, by providing real-time instructions and feedback through telemedicine apps, the system's AI analyzes live image quality and gives suggestions on how to move and position the ultrasound's transducer to improve clarity. This can help more front-line hospital staff conduct examinations on their own, reducing exposures of sonography personnel. Philips Lumify and Butterfly IQ, the largest handheld device manufacturers, introduced a tele-ultrasound system that streams the images live over a video call or over a smartphone or tablet.

*Preliminary analysis:* We have introduced a computer-based system that captures changing patterns in pleural lines associated with COVID-19 by means of a quantitative analysis of lung ultrasound [69]. The analysis only requires a minimal input from the user, who selects a seed region and then validates the segmentation Figure 1A. The use of the algorithm requires minimal training. Quantitative lung features describing pleural line thickness and margin morphology including tortuosity, projected intensity deviation, and nonlinearity were evaluated Figure 1B. The use of the algorithm facilitated high accuracy in detecting COVID-19 and improved the diagnostic performance of LUS Figure 2.

In the current study, we compared the performance of five users with different experiences ranging from no clinical experience to a highly experienced user to analyze the ultrasound images using the algorithm. The users' backgrounds are summarized in Table 1.

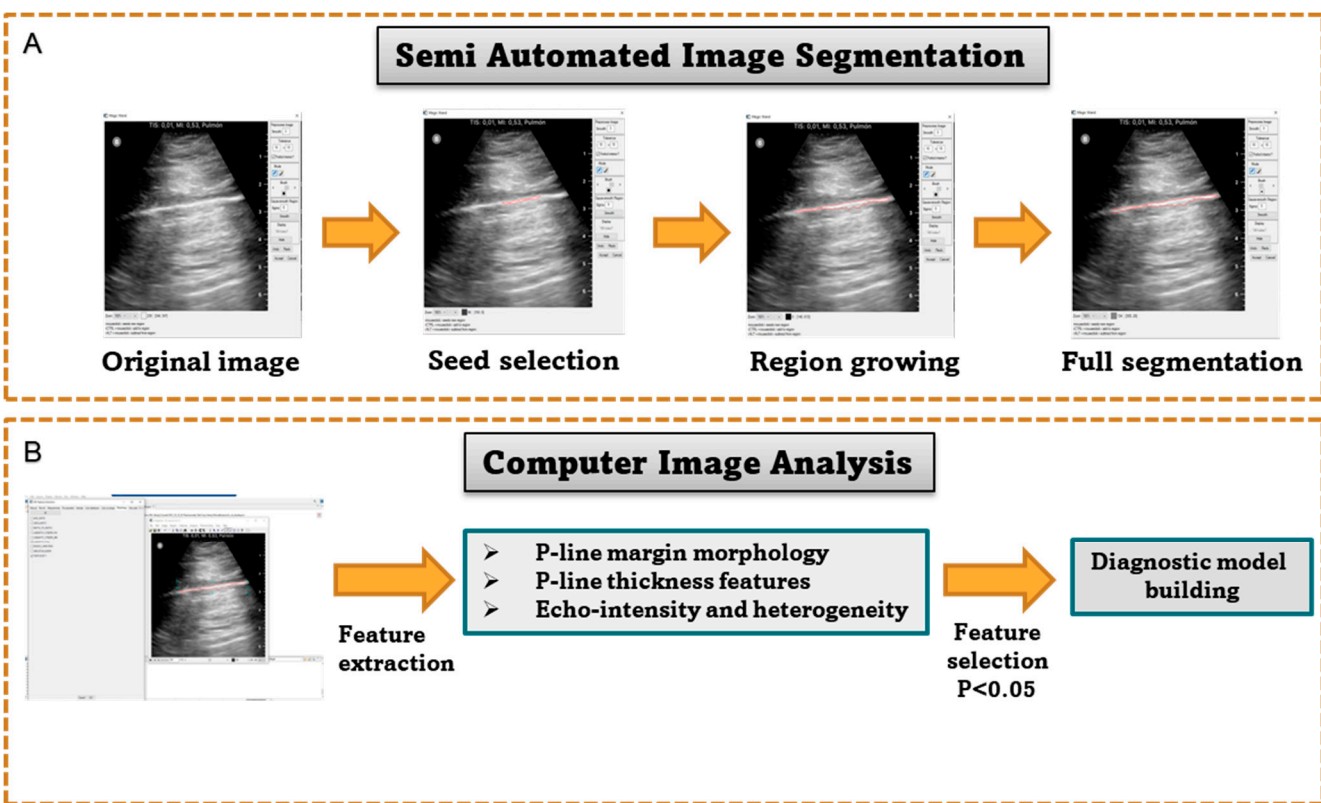

**Figure 1.** Schematic showing the computer-based image analysis steps. (**A**) demonstrates the semi-automated image segmentations steps. (**B**) shows the feature extraction and selection step towards building the diagnostic model.

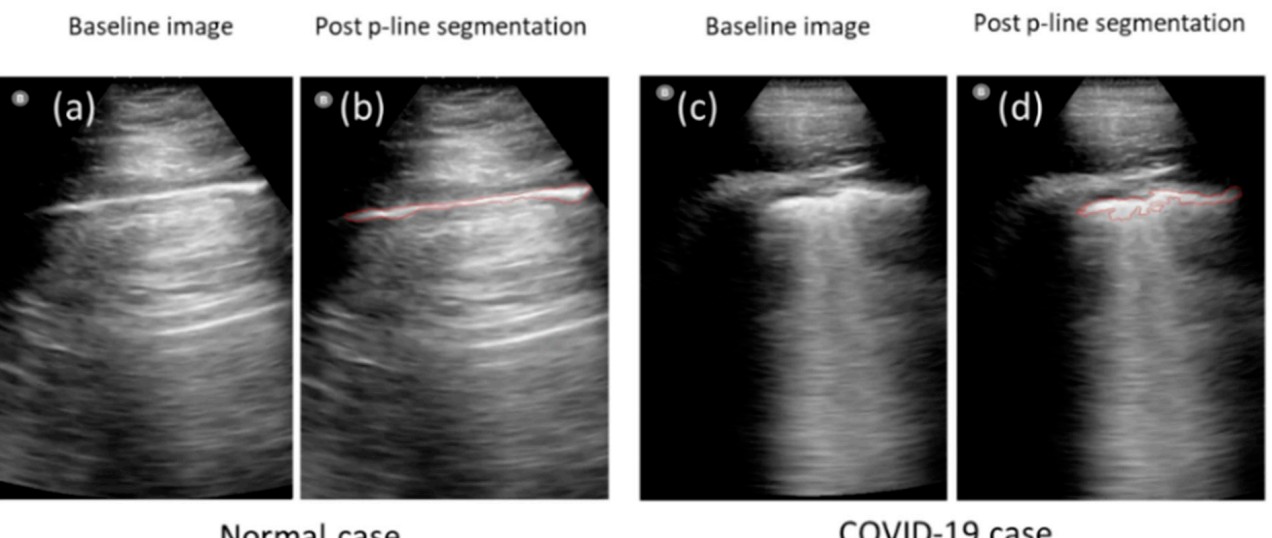

**Figure 2.** Demonstrates the high accuracy of semi-automated segmentation method of pleural lines. The left panels (**a**,**b**) show an example of a normal case, while panels (**c**,**d**) show a confirmed COVID-19 case, where p-lines in both cases were detected accurately by semi-automated segmentation.

**Table 1.** The clinical and educational backgrounds of AI-analysis tool users.

|  | Years of Clinical Experience | Experience in Diagnostic Methods | Experience with Ultrasound Technologies | Educational Background |
|---|---|---|---|---|
| User 1 | None | >30 years | >30 years | PhD |
| User 2 | >5 years | >5 years | >5 years | MD |
| User 3 | >5 years | >1 years | >1 years | MD |
| User 4 | None | >20 years | >20 years | PhD |
| User 5 | 10 years | 10 years | 10 years | MD |

The images were received for analysis anonymized and without patient-related information. Of twenty B-mode ultrasound images, ten images were acquired from COVID-19 patients, and another ten independent images were acquired from normal cases. The images were acquired using a portable ultrasound device (Butterfly IQ) using a specially designed lung-imaging setting. The lung preset has been optimized for demonstration and sliding of the pleural surface, as well as A-lines and B-lines throughout the 15 cm field. The preset transforms from a curvilinear array to a linear array when the depth is reduced to 7 cm or less. This optimizes near-field lung sliding dynamics as well as pleural artifacts. The same settings were used for acquiring all images used in the analysis. The patients' informed consent for performing image analysis on anonymized images was exempted by the University of Pennsylvania's institutional review board. In respect of the imaging data, all the methods employed were in accordance with relevant guidelines and regulations. The experimental protocol of the retrospective study was approved by the University of Pennsylvania's institutional review board. We performed a quantitative analysis that utilized image segmentation and feature extraction using IDL software (version 8.5, L3Harris Geospatial, Boulder, CO, USA) [69]. Training and guidance in image analysis were provided by the user with the highest experience to other users. To segment the pleural lines, we developed a semi-automatic segmentation tool called a "wand". This tool utilizes a basic form of "region growing", where the algorithm automatically grows the region by selecting pixel seeds with similar grayscale values within a tolerance range of ±10 gray levels (default value). The user's input is minimal, requiring only clicking and validating the segmentation and making corrections when necessary. After detecting the pleural line, the software extracts quantitative features that describe the depth (thickness), margin morphology, brightness, and heterogeneity [69,70]. The thickness parameters measure the change in the pleural line's horizontal depth. Margin morphology features, including tortuosity, projected intensity deviation (PID), and nonlinearity, measure irregularities in the pleural line's margin shape.

The results for the different users are shown in Table 2. The AUC (area under the ROC curve) as a metric for binary classification models based on the ROC (receiver operating characteristic) curve was used for the discrimination of the groups, normal vs. COVID-19. A strong performance was observed for the different users with the AUC ranging between 0.84 and 0.99. For users 1, 2, and 5, who had long experience with LUS imaging, the results showed that five features from seven were significantly different between COVID-19 and normal, ($p < 0.05$). The performance for these users ranged between 0.96 and 0.99. For user 3, who was a student familiar somewhat with LUS, the results showed that five features were also significant between normal and COVID-19 cases with a performance of the AUC of 0.93. For user 4, who was the least experienced user, the performance was respectable, with the AUC being 0.84.

**Table 2.** Feature classification and diagnostic performance of quantitative ultrasound features by different users.

| | | Thickness | Thickness Variation | PID | Nonlinearity | Tortuosity | Echo Intensity | Echo Heterogeneity | Overall Performance (AUC) |
|---|---|---|---|---|---|---|---|---|---|
| User 1 | COVID-19 | 5.20 | 0.23 | 2.68 | 0.25 | 1.54 | 186.86 | 18.45 | |
| | Normal | 1.80 | 0.06 | 0.70 | 0.81 | 1.04 | 195.06 | 14.69 | 0.96 |
| | *p*-value | 0.04 | 0.04 | 0.01 | 0.01 | 0.01 | 0.47 | 0.12 | |
| User 2 | COVID-19 | 4.62 | 1.97 | 2.45 | 0.20 | 1.46 | 184.79 | 20.96 | |
| | Normal | 1.41 | 0.36 | 0.49 | 0.90 | 1.01 | 197.41 | 18.60 | 0.98 |
| | *p*-value | 0.00 | 0.00 | 0.00 | 0.00 | 0.00 | 0.30 | 0.43 | |
| User 3 | COVID-19 | 6.03 | 0.25 | 2.45 | 0.23 | 1.33 | 129.20 | 34.46 | |
| | Normal | 2.51 | 0.06 | 0.59 | 0.74 | 1.08 | 137.74 | 32.38 | 0.92 |
| | *p*-value | 0.03 | 0.00 | 0.00 | 0.00 | 0.07 | 0.20 | 0.53 | |
| User 4 | COVID-19 | 4.69 | 0.67 | 1.62 | 0.23 | 1.51 | 202.21 | 13.91 | |
| | Normal | 1.10 | 0.85 | 1.84 | 0.79 | 1.16 | 212.31 | 11.50 | 0.84 |
| | *p*-value | 0.00 | 0.69 | 0.84 | 0.04 | 0.02 | 0.51 | 0.06 | |
| User 5 | COVID-19 | 5.84 | 1.40 | 3.03 | 0.13 | 1.10 | 167.21 | 34.62 | |
| | Normal | 2.16 | 0.44 | 0.93 | 0.78 | 1.04 | 182.48 | 27.56 | 0.99 |
| | *p*-value | 0.00 | 0.01 | 0.01 | 0.00 | 0.04 | 0.22 | 0.04 | |

Using the computer-based analysis tool improved the consistency of the readers' performance. On average, the interclass agreement was $0.83 \pm 0.12$, indicating excellent agreement. In comparison, the interclass correlation coefficient (ICC) for clinical assessment without the algorithm was 0.77, indicating good agreement. The agreement between readers was comparable and even surpassed the ICC without the algorithm. These results align with the previous literature, where AI algorithms achieved results that are comparable to human performance [41].

The automated segmentation methods that were performed in this study can be easily integrated into the operation of a scanner for real-time bedside assessment. In addition, fully automated machine learning segmentation could be optimized and performed. This technology, when implemented successfully in clinical practice, will increase confidence in diagnosis, especially in low-resource communities around the globe that lack experience in lung ultrasound. Automated systems for lung ultrasound data analysis are an active area of research, and there have been several recent advancements in this field. The automated segmentation of the pleural line is a crucial step in the automated analysis of LUS images, as it provides accurate measurements of pleural line changes, which are important indicators of lung pathology. There have been few recent studies that have focused on developing automated segmentation methods for the pleural line in lung ultrasound images. One common approach is to use deep learning algorithms, such as convolutional neural networks (CNNs), to segment the pleural line automatically. For example, a study published in 2020 proposed a fully automated deep learning algorithm for segmenting the pleural line in LUS images [71]. The algorithm used a U-Net architecture, a type of CNN commonly used for image segmentation, and achieved high accuracy in segmenting the pleural line. Another study published in 2021 proposed a segmentation method based on the combination of a deep learning algorithm and an active contour model [72]. The proposed method achieved high accuracy in segmenting the pleural line in both healthy and diseased lungs and demonstrated promising results in a clinical setting. Furthermore, there have also been studies that have focused on developing methods for correcting errors in automated pleural line segmentation. For example, a study published in 2020 proposed a method for correcting errors in automated pleural line segmentation using a graph-based approach [73]. Overall, these studies demonstrate that automated segmentation of the pleural line using deep learning algorithms is a promising approach

for analyzing LUS images. In our studies, we evaluated a simpler method of segmentation and feature extraction that can be easily implemented in clinical settings.

The present study has certain limitations, primarily due to its retrospective nature and the small sample size of the image data. To establish the method's validity, future prospective studies with larger datasets are necessary. Additionally, the variability resulting from the selection of a single image from the clip can be addressed through advancements in automated segmentation and region selection techniques, offering more accurate and reliable results. Further research is warranted to develop improved methods for analyzing the pleural line, particularly considering the functional aspects such as lung sliding motion, in addition to structural changes. Incorporating features that capture both functional and structural aspects could enhance disease detection. Furthermore, to enhance the safety and effectiveness of AI tools for remote monitoring with POCUS, several critical safeguards must be in place. These include rigorous validation and testing to ensure the system's accuracy, human oversight to review and confirm AI-generated findings, continuous monitoring and feedback mechanisms for ongoing improvement, comprehensive education and training for users with limited experience, clear communication of the AI tool's confidence levels, well-defined protocols for risk mitigation in cases of false positives or negatives, accessible user support channels, regular system updates, established lines of accountability, and a commitment to ethical considerations that prioritize patient well-being above all else. These safeguards collectively serve to mitigate the risks associated with false positives and false negatives, fostering trust in AI-driven healthcare solutions while prioritizing patient safety.

In developing the proposed system for remote monitoring with POCUS, valuable lessons are learned from existing self-monitoring modalities such as blood-sugar and blood-pressure monitoring. Key principles include designing a user-friendly interface with clear instructions, automating data capture to minimize user error, providing real-time feedback and data visualization, implementing alerts and notifications for critical findings, emphasizing user education and training, integrating remote monitoring and telehealth capabilities, prioritizing data privacy and security, ensuring accessibility for all users, and maintaining a feedback-driven approach for continuous improvement. By incorporating these lessons, the authors can enhance the usability, effectiveness, and safety of their system, empowering patients with limited experience to actively participate in their healthcare while maintaining data accuracy and reliability.

*Future opportunities for AI:* Another form of AI that holds incredible promise for healthcare involves augmented and virtual reality (AR/VR) technology [74–77]. Both are quickly bringing forth possibilities that were never previously imagined. The concept of the metaverse, for example, has great potential for an entirely new approach to enhance telehealth solutions by reducing the distance between the patients and their providers. It can elevate virtual care from a 2D to a 3D experience, changing the way patients interact with healthcare providers in a fundamental way. Facebook's recent transformation to "Meta" is an ode to the company's foresight and vision that the metaverse is the next frontier in technological innovation. Another example is Microsoft Mesh, a "holoportation" and mixed reality platform that aims to make digital connections lifelike and enable new ways to remotely teach, learn, and perform tasks virtually. This comes in line with Microsoft's Hololens technology that has been investigated for efficacy in remote medical care in surgical and nonsurgical case studies. Companies are also investing in augmenting the telemedicine experience through AR/VR for more friendly and safe patient experiences.

In recent studies focused on COVID-19 diagnosis, researchers have explored the potential of VR and AR immersive technologies [78–80]. These investigations have sought to integrate CT imaging with VR technology to create an automated system capable of navigating 3D visualizations. The objective has been to develop a VR platform specifically designed for visualizing and interacting with 3D representations of the lungs and infected regions. By leveraging AR/VR technology, it is envisioned that remote monitoring of

COVID-19 patients with POCUS can be further facilitated, particularly in terms of education, training, and guidance for patients and other non-trained personnel.

## 6. Conclusions

Remote monitoring of COVID-19 patients in the comfort of their homes holds significant promise, made feasible through the integration of AI-supported teleguidance systems. This innovative approach POCUS technology, even in the hands of untrained individuals, including the patients themselves. In this paper, we have elucidated the mechanisms for effectively training patients to adeptly utilize teleguided portable ultrasound devices for self-monitoring and reporting changes pertaining to the progression of the disease. Our preliminary analysis has yielded encouraging results, revealing that individuals with limited experience can achieve a remarkable level of accuracy in detecting COVID-19 when assisted by AI. The implications of this breakthrough are profound; if implemented successfully on a broader scale, it stands to substantially reduce the exposure of healthcare workers to the virus and curtail the risk of unnecessary disease transmission. In essence, this AI-driven telemonitoring system not only empowers patients to actively participate in managing their health but also serves as a vital tool in our collective efforts to mitigate the spread of COVID-19, ultimately safeguarding public health and healthcare resources.

**Author Contributions:** L.R.S., A.H. and G.D. conceptualized the study. Y.T.-C. acquired the data. L.R.S., M.A.-H., T.W.C. and C.M.S. analyzed and interpreted the data. L.R.S. and A.H. wrote the manuscript. This article contains original data. All authors have read and agreed to the published version of the manuscript.

**Funding:** This research received no external funding.

**Institutional Review Board Statement:** The study did not require ethical approval. The LUS images were received anonymized for analysis with no patient information.

**Informed Consent Statement:** Not applicable.

**Data Availability Statement:** The datasets generated and/or analyzed during the current study are not publicly available due the ongoing nature of the research study but are available from the corresponding author on reasonable request.

**Conflicts of Interest:** The authors declare no conflict of interest.

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
