# Peer review of "Can Artificial Intelligence Aid Diagnosis by Teleguided Point-of-Care Ultrasound? A Pilot Study for Evaluating a Novel Computer Algorithm for COVID-19 Diagnosis Using Lung Ultrasound"

_ai, doi:10.3390/ai4040044_

Round 1

Reviewer 1 Report

This communication reviewed the need of teleguided POCUS during COVID-19 pandemic, and summarized some of the promising findings.  The manuscript is overall well written. However, I think some important aspects of teleguided POCUS have been overlooked.  There is no argument that AI can improve the diagnosis. The manuscript spent a lot of space to demonstrate that AI can automatically process the provided images. But in a teleguided setting with a person who is less/no experience in performing US, the largest concern is whether the person can produce images with adequate qualify for the subsequent analysis. So, to me, a more important question at this time is that how to use AI to help a novice user producing good US images.  To do so, efforts in both hardware and software will be needed. I suggest the author to more focus on these major challenges.

The study presented data from 5 users of different experience levels.  There are a lot of details about the users were missing. How is the experience level determined? Are there any guidance/help provided during performing US? If so, by real people or AI? How do you group the users and what is the statistics for the study? 

Author Response

This communication reviewed the need of teleguided POCUS during COVID-19 pandemic, and summarized some of the promising findings.  The manuscript is overall well written. However, I think some important aspects of teleguided POCUS have been overlooked.  There is no argument that AI can improve the diagnosis. The manuscript spent a lot of space to demonstrate that AI can automatically process the provided images. But in a teleguided setting with a person who is less/no experience in performing US, the largest concern is whether the person can produce images with adequate qualify for the subsequent analysis. So, to me, a more important question at this time is that how to use AI to help a novice user producing good US images.  To do so, efforts in both hardware and software will be needed. I suggest the author to more focus on these major challenges.

Response: We thank the reviewer for the valuable comment. We agree that the main goal is to ensure that the AI can help users with no or less experience in image acquisition. In the manuscript, we discuss the possible benefits of AI tools when implemented in ultrasound systems, helping users with lesser experience to acquire images with high accuracy. The purpose of the example tool that we evaluate in this study is to show that similar AI tools can help in improving automated detection of normal and abnormal findings even with users of lesser experiences. This tool uses semi-automated detection methods and needs minimal input to detect. The current tool, if validated on larger data set and more users, can be implemented on the portable device and hence help users with less experience in imaging by automated segmentation. More details were added to the manuscript to ensure that the purpose of assessing the tool and the future goals are clear to the readers, please see lines 51-61.

The study presented data from 5 users of different experience levels.  There are a lot of details about the users were missing. How is the experience level determined? Are there any guidance/help provided during performing US? If so, by real people or AI? How do you group the users and what is the statistics for the study? 

Response: We thank the reviewer for the value comment. We agree that some details are missing in this regard. Accordingly, they were added to the manuscript. Please see table 1.

Training and guidance in image analysis was provided by the user with the highest experience to other users. These details have been added to the manuscript, please see lines 224-225.

Reviewer 2 Report

Please refer to the attachment for specific modifications.

 Grammar needs polishing.

Author Response

Reviewer 2:

This paper proposes the use of remote guided POCUS supported by artificial intelligence

technology to realize the remote monitoring of patients with new coronary pneumonia by

inexperienced personnel. The authors used a computer-based system for lung ultrasound detection

of COVID-19 to evaluate the performance of users in different clinical backgrounds. But there are

still following problems:

(1) The contribution of this paper is not clear. The authors should further innovate and

contribute to this paper.

We thank the reviewer for the valuable comment. In the manuscript, we discuss the possible benefits of AI tools when implemented in ultrasound systems, helping users with lesser experience to acquire images with high accuracy. The purpose of the example tool that we evaluate in this study is to show that similar AI tools can help in improving automated detection of normal and abnormal findings even with users of lesser experiences. This tool uses semi-automated detection methods and needs minimal input to detect. The current tool, if validated on larger data set and more users, can be implemented on the portable device and hence help users with less experience in imaging by automated segmentation. More details were added to the manuscript to ensure that the purpose of assessing the tool and the future goals are clear to the readers, please see lines 51-61.

(2) The paper structure is very confusing.

Response: we thank the reviewer but again the nature and structure of the paper as a commentary article is different than an original research article.

(3) The experiment is too weak to prove the conclusion. More experiments and discussions

should be added.

Response: We thank the reviewer again, the purpose of using this preliminary data as an example for potential advantages of AI in improving the diagnosis for users with different levels of experience. Larger studies will be needed for validation.  This has been clarified more in the manuscript, please see lines 51-61.

(4) The authors should add the last five years of algorithms for comparative analysis.

We thank the reviewer, more details about similar algorithms was added to the manuscript.

(5) Many references are very old.

Response: We thank the reviewer for the comments, as requested recent references were added to the manuscript.

(6) The expression of the conclusion needs to be improved, and there is a lack of prospect for

future work. Also, the authors should add a limitations analysis.

Response: conclusions section has been updated.

Reviewer 3 Report

Summary:

This study is a retrospective evaluation of an AI assisted COVID-19 detection system. Users with a range of clinical experience used this system to evaluate ultrasound images. The authors evaluated the performance of the users, in the context of their experience, to assess the accuracy and congruence of the diagnosis presented by the users.

Comments:

Line numbers are missing from the manuscript.

The author affiliations seem to be in different font sizes. I don't know if this is just in my version of the document.

I incorrectly assumed User 1 had the least training, and User 5 was the most experienced, based on the initial wording of the user population. Can the authors include a table describing the characteristics of the 5 users? We don't need demographics, but information regarding years of clinical experience, experience with diagnostic methods, experience with ultrasound technologies etc., can help quantify the differences between the users. 

The authors have used AUC as a measure of performance. Could the authors please expand on how the AUC is calculated? How is the AUC a better measure of performance than something like root mean squared error(RMSE) for continuous variables?

Can the authors expand on the impact of false positives and false negatives on overall care when using such methods? Since the system is targeted for use by people with limited experience, what safeguards should be in place to avoid false positive/negatives (whichever is more dangerous)?

The authors describe their image quantification methods on page 5-6. I am not qualified to judge the validity of this method, but is this a standard way of quantifying ultrasound images? Could the authors expand on this for users like me.

There are several diagnostic modalities like blood sugar monitoring and blood pressure monitoring, where less experienced users like patients are already self-monitoring their health using smart devices. What lessons from such methods can be applied to the authors proposed system?

References need to be reformatted based on journal guidelines.

Author Response

Reviewer 3: This study is a retrospective evaluation of an AI assisted COVID-19 detection system. Users with a range of clinical experience used this system to evaluate ultrasound images. The authors evaluated the performance of the users, in the context of their experience, to assess the accuracy and congruence of the diagnosis presented by the users.

Comments:

Line numbers are missing from the manuscript.

Response: we thank the reviewer for the comment. As requested, line numbers were added.

The author affiliations seem to be in different font sizes. I don't know if this is just in my version of the document.

Response: thanks again to the reviewer, the font was corrected accordingly.

I incorrectly assumed User 1 had the least training, and User 5 was the most experienced, based on the initial wording of the user population. Can the authors include a table describing the characteristics of the 5 users? We don't need demographics, but information regarding years of clinical experience, experience with diagnostic methods, experience with ultrasound technologies etc., can help quantify the differences between the users. 

Response: As requested, a table describing the user backgrounds was added. Please see table 1.

The authors have used AUC as a measure of performance. Could the authors please expand on how the AUC is calculated? How is the AUC a better measure of performance than something like root mean squared error (RMSE) for continuous variables?

Response: We thank the reviewer for the valuable comments. Root Mean Squared Error (RMSE) is a widely used metric to evaluate the accuracy of a predictive model, particularly in the context of regression analysis. As we don’t have a predictive model but rather, we just compared the performance of individual features in differentiating normal from the diseased, therefore our choice was to use AUC (Area Under the ROC Curve) as a metric for binary classification models based on the ROC (Receiver Operating Characteristic) curve. AUC summarizes this performance by measuring the area under the ROC curve, with higher values indicating better discrimination capability of the groups, normal vs COVID-19. More details about AUC were added to the manuscript, please see lines 236-238.

Can the authors expand on the impact of false positives and false negatives on overall care when using such methods? Since the system is targeted for use by people with limited experience, what safeguards should be in place to avoid false positive/negatives (whichever is more dangerous)?

Response: We thank the reviewer for the important comment. As requested, more details have been added to the manuscript, please lines 292-305

The authors describe their image quantification methods on page 5-6. I am not qualified to judge the validity of this method, but is this a standard way of quantifying ultrasound images? Could the authors expand on this for users like me.

Comment: The standardization of image quantification methods in ultrasound can vary depending on the specific application and the field of study. In some cases, there are standardized techniques and measurements, while in others, methods may be more research-oriented and subject to variation. In established clinical applications, there is more standardization compared to research-oriented or emerging areas of ultrasound imaging ( like what we are proposing) where methods may be less standardized and subject to variation.

There are several diagnostic modalities like blood sugar monitoring and blood pressure monitoring, where less experienced users like patients are already self-monitoring their health using smart devices. What lessons from such methods can be applied to the authors proposed system?

Response: This is a very nice insight. We added a paragraph related to the subject, please lines 305-315.

Round 2

Reviewer 1 Report

Nothing

Reviewer 2 Report

There are many studies on auxiliary diagnosis. It is recommended that the author add more latest studies, such as 10.1109/BIBM55620.2022.9995391.

Improvements can be made